# Combined Transcranial Direct Current Stimulation and Functional Electrical Stimulation for Upper Limbs in Individuals with Stroke: A Systematic Review

**DOI:** 10.3390/neurolint17060089

**Published:** 2025-06-09

**Authors:** Alfredo Lerín-Calvo, Juan José Fernández-Pérez, Raúl Ferrer-Peña, Aitor Martín-Odriozola

**Affiliations:** 1Grupo de Investigación en Neurociencias Aplicadas a la Rehabilitación (GINARE), 28923 Alcorcón, Spain; alfredo.lerin@lasallecampus.es; 2Departamento de Fisioterapia, Centro Superior de Estudios Universitarios La Salle, Universidad Autónoma de Madrid, Aravaca, 28023 Madrid, Spain; raul.ferrer@salud.madrid.org; 3Grupo de Investigación Clínico Docente Sobre Ciencias de la Rehabilitación (INDOCLIN), CSEU La Salle, UAM, Aravaca, 28023 Madrid, Spain; 4Centro de Salud Entrevías, Gerencia Asistencial de Atención Primaria de la Comunidad de Madrid, 28023 Madrid, Spain; 5Fesia Technology, 20009 Donostia-San Sebastián, Spain; aitor.martin@ehu.eus; 6Fesia Clinic, 20014 Donostia-San Sebastián, Spain; 7Physiology Department, University of the Basque Country (UPV/EHU), 48940 Leioa, Spain

**Keywords:** tDCS, functional electrical stimulation, stroke, neuromodulation, neurorehabilitation, motor recovery

## Abstract

Background: Transcranial direct current stimulation (tDCS) and functional electrical stimulation (FES) are established interventions to enhance upper limb motor function (ULMF) in people with stroke (PwS). However, evidence supporting their combined use remains limited and inconsistent. This systematic review aims to evaluate the effectiveness of combined tDCS and FES for improving ULMF, activity, and participation in PwS. Methods: A systematic search was conducted across MEDLINE, CINAHL, SPORTDiscus, CENTRAL, SCOPUS, and Web of Science from inception to December 2024. Randomized and controlled clinical trials (RCTs) involving adults (≥18 years) with acute, subacute, or chronic stroke undergoing combined tDCS and FES interventions were included. Methodological quality was assessed with the PEDro scale, and risk of bias was evaluated using the Cochrane RoB2 tool. A qualitative synthesis was performed employing a five-level evidence grading system. Results: Five RCTs involving 148 participants (mean age range: 50.6–61.2 years; 26% female) were included. Stroke chronicity ranged from 7.6 days to 27.5 months post-onset. Four studies reported significant ULMF improvements with the combined intervention. However, activity and participation outcomes were inconsistently assessed, and results remained inconclusive. Methodological quality varied, with one study rated as excellent, two as good, one as fair, and one as poor. The risk of bias was rated high or with concerns in four out of five studies. Conclusions: Based on qualitative synthesis, moderate-level evidence supports the combined use of tDCS and FES for improving ULMF in PwS. However, high variability in protocols, small sample sizes, and the increased risk of bias in most studies limit the strength of these conclusions. Standardized protocols and larger high-quality RCTs are needed to confirm the effectiveness of this combined intervention.

## 1. Introduction

Stroke remains the second leading cause of death and the third leading cause of disability-adjusted life years globally, with an estimated economic burden of USD 721 billion [1]. Stroke is characterized by the abrupt onset of neurological deficits resulting from inadequate cerebral blood flow due to ischemic (87%) or hemorrhagic (13%) injury [2]. A prevalent sequela of stroke is paresis, which affects 49–57% of people with stroke (PwS), leading to upper limb (UL) weakness [3,4]. Approximately 60% of PwS with severe impairment and 30% with moderate impairment report having a non-functional UL 6 months post-stroke, and this finding suggests that the UL dysfunction has a significant impact on their activities of daily living [5].

Current evidence indicates that functional rehabilitation targeting the UL can be effective in PwS [6]. This approach aims to restore and maximize the functional capacity of existing movements by inducing adaptive neural plasticity [7,8]. Advances in electrical stimulation technologies have significantly contributed to achieving these neuroplasticity-related changes [9,10]. Consequently, electrical stimulation is the second most frequently applied rehabilitation technique among PwS within 1–3 months from stroke onset [11], and transcranial direct current stimulation (tDCS) and functional electrical stimulation (FES) are widely utilized as rehabilitation techniques.

As a non-invasive neuromodulation technique, tDCS employs a low, constant, direct current to modulate neuronal activity in the brain [12]. By applying a mild electrical current through electrodes on the scalp, tDCS can alter cortical excitability in targeted brain regions, influencing synaptic plasticity and neural network function [13]. It has received a level B recommendation for improving UL function and activities of daily living [14], with no significant differences between stimulation polarities [15]. Consequently, tDCS is becoming increasingly integrated into rehabilitation programs aimed at optimizing recovery for PwS.

FES is an electrical stimulation technique that targets the peripheral nervous system to induce muscle contractions that mimic functional movements, primarily via a transcutaneous approach [16]. FES facilitates motor recovery through two primary modes: (1) open-loop, which involves preprogrammed stimulation unaffected by patient feedback [17]; and (2) closed-loop, which adjusts stimulation in real time, based on feedback from physiological signals such as electromyography or electroencephalography [18]. Recent meta-analyses have shown that FES is superior to other electrical stimulation methods for improving UL function and can be combined with robotic-assisted training, brain–computer interfaces, or motor imagery [19,20].

Although the main contribution of FES to UL recovery is related to its effects on the peripheral nervous system and muscle due to an increase in motor unit recruitment [21], changes in the sensorimotor cortex have been observed after peripheral electrical stimulation [22]. In turn, the combination of central and peripheral stimulation appears to be an interesting line of treatment for neurological patients due to an increase in corticospinal excitability and its potential benefits for improved motor outcomes [23].

Despite the established benefits of FES and tDCS when used separately, their concomitant application is less common in clinical practice. To date, no systematic reviews have examined the combined use of these modalities. It is, therefore, imperative to conduct a comprehensive systematic review to elucidate the effects of this combination, identify optimal parameters and dosages, and ultimately enhance rehabilitation outcomes for PwS. This review will address a critical gap in the literature and provide valuable insights into how these techniques can be effectively integrated to maximize their therapeutic benefits.

## 2. Materials and Methods

This systematic review was conducted according to the recently updated PRISMA (Preferred Reporting Items for Systematic Reviews and Meta-analysis) 2020 statement [24]. This review has been registered in PROSPERO (ID: CRD42024547726).

### 2.1. Eligibility Criteria

The selection criteria used in this review were based on clinical and methodological aspects, such as population, intervention, control, outcomes, and study design. The criteria were thus based on attempting to answer the PICOS question: In terms of population (P), intervention (I), control (C), dependent variable(s) or result(s) of interest (O), and study type (S), is the combination of transcutaneous direct current stimulation and functional electrical stimulation effective for improving UL motor function in PwS [25]?

### 2.2. Population

The participants included in the reviewed studies were PwS, aged 18 years or older, and presenting UL deficits, regardless of the time since stroke onset (acute, subacute, or chronic).

### 2.3. Intervention and Comparison

The studies included were randomized controlled trials (RCTs) that included tDCS plus FES in combination with or without other techniques, such as task-oriented training. The control group received either tDCS, FES alone, or sham tDCS combined with FES. The authors considered FES to be an application of electrical impulses to nerves or muscles to elicit contractions supporting the execution of functional task-oriented movements.

### 2.4. Outcome Measurements

Variables studied to test the results, and the effects of the treatments were UL motor function and activity. The Fugl-Meyer Assessment of the Upper Extremity (FMA-UL) was selected to measure motor function. To assess UL activity, the Action Research Arm Test (ARAT), Motor Activity Log (MAL), or Wolf Motor Function Test (WMFT) were employed.

### 2.5. Study Type

RCTs and controlled clinical trials (CCTs) with a parallel-arm design were eligible for this review. No language or time filters were applied.

### 2.6. Search Strategy

Two independent reviewers searched scientific articles, thus generating an agreement for the initial selection of studies after which discordances were sought. The search for scientific articles was conducted using the following databases: MEDLINE via PubMed, CINAHL plus SPORTDiscus via EBSCO, Cochrane Central Register of Controlled Trials (CENTRAL), SCOPUS, and Web of Science. The Mesh terms used for the search strategy were as follows: “transcranial direct current stimulation,” “electrical stimulation therapy,” “paresis,” “stroke,” and “upper extremity.” The search for scientific articles included a backward citation search ending in December 2024. The search strategy used for each database is reflected in Appendix A.

### 2.7. Selection Criteria and Data Extraction

The following inclusion criteria were applied to select studies for this review: (1) RCTs or CCTs with a parallel-arm design; (2) participants aged ≥18 years diagnosed with stroke and presenting UL motor deficits; (3) interventions including the combined application of tDCS and FES, with or without additional therapies; (4) comparators including tDCS alone, FES alone, or sham stimulation combined with FES; (5) outcomes including at least 1 validated measure of UL motor function or activity; and (6) full-text availability.

The exclusion criteria were as follows: (1) studies with non-stroke populations or without UL motor impairment; (2) interventions involving other forms of electrical or brain stimulation not meeting the defined criteria; (3) studies not reporting relevant clinical outcomes related to motor function or activity; and (4) abstracts, conference proceedings, protocols, or non-peer-reviewed publications. The decision to include only randomized controlled trials (RCTs) and controlled clinical trials (CCTs) was based on the need to ensure a high level of internal validity and minimize bias in assessing the combined effects of tDCS and FES.

An initial analysis was performed by 2 independent reviewers (A.L.C. and A.M.O.) who assessed the relevance of the RCTs in relation to the research objective. The first analysis was performed based on the information extracted from the title and abstract of the study. The full text was accessed if there was no consensus or if the abstracts did not contain the necessary information. The Rayyan platform was employed during the selection process to identify and remove duplicate records.

In the second phase of the analysis, the full text of the articles was read to check which of them met the inclusion criteria. Moreover, the authors were contacted to see whether information about interventions or results were missing. Differences between reviewers were resolved by a third reviewer. Information regarding participant characteristics and each outcome specified in the inclusion criteria were systematically extracted and are presented in tabular form.

### 2.8. Risk of Bias Assessment

Two authors (A.L.C. and A.M.O.) independently assessed the risk of bias in each study, using version 2 of the Cochrane Risk of Bias tool for randomized trials (ROB2) in its 2 versions: parallel group trials and crossover trials. The ROB2 tool includes signaling questions to evaluate specific domains to help judge the randomized trial’s risk of bias, with possible answers comprising “yes,” “probably yes,” “probably no,” “no,” or “no information.” The risk of bias is therefore judged as “low,” high,” or “unclear.” This tool covers 6 domains: the randomization process (selection bias); deviations from intended interventions (performance bias); missing outcome data (attrition bias); measurement of the outcome (detection bias); selection of the reported results (reporting bias); and overall bias [26]. Disagreements between reviewers were resolved by consensus with a third reviewer.

Inter-rater agreement (inter-rater reliability) was measured with Cohen’s kappa coefficient (κ) as follows: (1) κ > 0.7 indicated a high level of inter-rater agreement; (2) κ = 0.5–0.7 indicated a moderate level of agreement; and (3) κ < 0.5 indicated a low level of agreement) [27].

### 2.9. Methodological Quality Assessment

The two independent reviewers (A.L.C. and A.M.O.) assessed the methodological quality of the randomized trials, assessing each of the selected studies based on the PEDro scale developed by Barton et al., a scale demonstrated to be a valid and reliable tool for assessing the methodological quality of systematic reviews. With a total of 13 items, each worth 2 points (with “yes” scoring 2; “in part” scoring 1; and “no” scoring 0), the maximum possible score was 26. A high-quality cut-off of 20 or more points was provided by the developers. The exclusion and keyword criteria were modified to better evaluate the selected systematic reviews of this study [28]. Disagreements between reviewers were resolved by consensus with a third reviewer.

Inter-rater agreement (inter-rater reliability) was measured with Cohen’s kappa coefficient (κ) as follows: (1) κ > 0.7 indicated a high level of inter-rater agreement; (2) κ = 0.5–0.7 indicated a moderate level of agreement; and (3) κ < 0.5 indicated a low level of agreement) [27].

### 2.10. Qualitative Analysis

To evaluate the qualitative analysis of the results, the evidence was classified into 5 grades depending on the methodological quality of the studies, as follows [29]. This qualitative analysis has been used in several systematic reviews [30,31,32].

Strong evidence: Provided by statistically significant findings of outcome measures in at least 2 high-quality RCTs.Moderate evidence: Provided by statistically significant findings of outcome measures in at least 1 high-quality RCT and at least 1 low-quality RCT and/or 1 high-quality CCT.Limited evidence: Provided by statistically significant findings of outcome measures in at least 1 high-quality RCT and/or at least 2 high-quality CCTs (in the absence of a high-quality RCT).Indicative findings: Provided by statistically significant findings of outcome measures in at least 1 high-quality CCT and/or low-quality RCT (in the absence of high-quality RCTs) and/or 2 studies of a nonexperimental nature of sufficient quality (in the absence of RCTs and CCTs).No or insufficient evidence: Cases in which the results of eligible studies did not meet the criteria for one of the levels of evidence indicated above and/or in the case of conflicting results (statistically significant positive and statistically significant negative) between RCTs and CCTs or in the case of a lack of eligible studies.

## 3. Results

This study’s screening strategy is shown in Figure 1. Of the 598 studies, 441 were duplicates and 14 had the full text reviewed. Of these studies, five met the inclusion criteria and were included in this review. Although we attempted to contact the authors, the requested information could not be obtained.

### 3.1. Characteristics of the Study Population and Study Design

A total of 148 PwS were ultimately included in this systematic review. Some studies were excluded due primarily to an incorrect intervention (*n* = 6) [33,34,35,36,37]. Thirty-nine (26%) participants were female, the mean age ranged between 50.6 and 61.2 years, and the time since stroke onset varied from 7.6 days to 27.5 months. Sixty-five (44%) patients were affected in the right hemisphere, whereas the type of stroke was not reported in three out of the five included studies. All of the patients in the included studies had UL limitations due to stroke. The type of design used for the search was an RCT with a parallel-arm design, and only one study included a follow-up period of 4 months. Detailed information for each study characteristic can be found in Table 1.

### 3.2. Characteristics of the Interventions and Comparators

All of the studies applied interventions based on combined tDCS and FES, except for two trials [38,42], where the studies combined these techniques with occupational therapy and physical therapy, respectively. The comparator groups were tDCS plus occupational or physical therapy [38,42], sham tDCS plus FES [39,40], or FES alone [41].

Regarding tDCS parameters, the electrode location was the same in all the studies. The anode was placed over the ipsilesional primary motor cortex (M1), and the cathode was placed over the contralesional M1. The intensity ranged from 1.2 to 2.5 mA, and the session duration was between 20 and 30 min.

Concerning FES, the stimulation frequency was set between 20 and 40 Hz, with a pulse width ranging from 50 to 500 µs. The intensity threshold was different in each study, with some authors basing this parameter on the PwS’s preferences, whereas other authors established a fixed limit. A detailed description of each study intervention is shown in Table 2 and Table 3, with information related to the specific tDCS and FES parameters.

### 3.3. Outcomes Measured

The results of the included studies were systematically categorized according to the International Classification of Functioning, Disability, and Health (ICF) framework [43], distinguishing between outcomes related to function, activity, and participation.

All of the included studies evaluated UL function. The FMA-UE was employed by all of the studies included [38,39,40,41,42]. The smoothness and velocity of reaching tasks by kinematic analyses [39], handgrip and pinch strength [39,42], and neurophysiological measures (surface electromyography and motor evoked potentials) [40] were also the function-related outcomes measured.

Activity-related outcomes were assessed in three studies, highlighting the capacity to perform specific tasks. The ARAT [38], WMFT [40,42], and 9-hole peg test (NHPT) [42] were utilized. Three studies included an activity-related variable. Lastly, participation outcomes were assessed using the Chedoke Arm and Hand Activity Inventory (CAHAI) [42] and the Motor Activity Log (MAL), which were utilized by two of the included studies [38].

### 3.4. Methodological Quality and Risk of Bias Results

The methodological quality of the studies was assessed using the PEDro scale. Of the five studies, two demonstrated good quality (score of 6), one showed fair quality (score of 5), one was rated as poor (score of 3), and one exhibited excellent methodological quality (score of 9).

Table 4 shows the results of the assessment of the studies according to the PEDro scale. The two reviewers reported discrepancies in the assessment of one RCT; the discrepancy observed in this study [40], was for concealed allocation. Of the five studies included, three were rated as having some concerns regarding risk of bias, one was assessed as having a low risk of bias [39], and one presented a high risk of bias [38]. Conflicts were resolved by consensus through a third reviewer. The inter-rater reliability of the methodological quality assessment was high (κ = 0.950).

The risk of bias was assessed with the revised Cochrane risk-of-bias tool for randomized trials (RoB2) (Figure 2). The Domain with the highest risk of bias was the randomization process, and the domain with the lowest risk of bias was missing outcome data. The inter-rater reliability of the risk of bias assessment was high (k = 0.867). A summary of each domain is presented in Figure 2. No imputation methods were required, as none of the included studies reported missing data for the primary outcome measures.

### 3.5. Observed Effects

#### Upper-Limb Motor-Function-Related Outcomes

The within-group comparisons revealed that FMA-UE scores increased using FES and tDCS in four out of the five studies included [38,39,40,41,42]. Only one study found significant differences in tDCS plus occupational therapy over occupational therapy alone [38]. The maximum isometric strength increased in two studies that included a tDCS plus FES group [39,42]; nevertheless, one study reported an increase in hand grip strength only in the tDCS plus physical therapy group [42]. Concerning the kinematic analysis, mean returning velocity was increased in both the tDCS plus FES and sham tDCS plus FES groups, whereas only reaching velocity was increased in the tDCS plus FES group in one study [39].

In the between-group comparisons, four out of the five studies reported significant improvements, favoring the tDCS plus FES intervention group for at least one measure of UL function [38,39,40,41,42]. Specifically, three studies demonstrated a more pronounced increase in FMA-UE scores in the tDCS + FES group compared to the control group [40,41,42], and one study observed significant differences in kinematic analysis regarding mean reaching velocity and movement time, but not in FMA-UE [39]. Furthermore, the maximal isometric strength [39,42] and sEMG [40] exhibited significant improvements in the tDCS + FES group compared to controls.

### 3.6. Activity and Participation-Related Outcomes

Regarding activity-related outcomes, within-group comparisons revealed that two out of three studies found changes in WMFT only in the tDCS plus FES group [28,29], while the NHPT time decreased and the ARAT score improved in both groups [38,42]. The between-group comparisons indicated an increase in WMFT and a decrease in NHPT time in favor of the tDCS plus FES group [40,42].

For participation variables, Hyakutake et al. reported an increase in MAL scores in the tDCS plus occupational therapy group only [38], whereas Devi et al. reported a significant increase in the CAHAI in both groups [42]. The only significant differences between groups were found in the CAHAI, favoring the tDCS plus FES and physical therapy intervention versus tDCS plus physical therapy alone [42].

### 3.7. Qualitative Synthesis

There is moderate evidence for the application of combined tDCS and FES for improving UL motor function in PwS provided by two high-quality RCTs [39,41], one low-quality RCT [42], one high-quality CCT [40], and one low-quality CCT [38].

However, there is insufficient evidence to support an improvement in UL motor activity with the combination of tDCS and FES due to the mixed results obtained in one high-quality CCT [40], one low-quality RCT [42], and one low-quality CCT [38].

## 4. Discussion

This systematic review is the first to evaluate the combined effectiveness of tDCS and FES for UL deficits in PwS, showing moderate evidence of improved function compared to controls. These benefits arise from tDCS enhancing cortical excitability and FES targeting motor recruitment, with the synergy being most effective in chronic stroke recovery [44]. However, limited studies on activity outcomes restrict the ability to make broader conclusions.

Our results partially align with the previous literature, indicating that this synergy is most effective during the chronic stages of stroke recovery [44]. tDCS alone showed superior effects on control interventions for enhancing UL function, demonstrating evidence level B [45,46], whereas a recent meta-analysis regarding different FES applications has revealed an improvement in UL function superior to non-FES conventional therapies (physical and occupational therapy) [19]. A recent meta-analysis showed that FES or tDCS combined with OT yielded better results than OT alone in ul motor function (ULMF) and quality of life [44]. Furthermore, favorable ULMF effects were reported in four out of the five studies included, suggesting a consistent trend in support of the combined tDCS and FES intervention. However, activity and participation outcomes were assessed in only a few studies, limiting the ability to establish a clear pattern in these domains. The observed variability across studies could be explained by differences in participant characteristics (e.g., stroke stage, age, sex), variations in stimulation protocols (e.g., target muscles), and the overall methodological quality and risk of bias of the included trials. Therefore, the results should be interpreted with caution, and further well-designed studies are needed to confirm these effects.

Despite the promising effects of individual and combined tDCS and FES, prior studies report conflicting evidence when central and peripheral stimulations are combined, resulting in heterogeneous outcomes. Yang et al. found statistically significant differences in the FMA-UE when combining FES with other non-invasive brain stimulation techniques, such as repetitive transcranial magnetic stimulation (rTMS) [47]. Conversely, repetitive transcranial plus peripheral magnetic stimulation (rPMS) did not show significant differences compared to rTMS alone for UL function in PwS [48]. An RCT using theta burst stimulation plus rPMS exhibited effects superior to theta burst stimulation and sham rPMS in the ARAT grasp domain, whereas no significant differences were found in total ARAT or FMA scores [49]. These inconsistencies in effects between our findings with tDCS plus FES and other stimulation combinations could reflect various factors. One important consideration is the lack of significant additive effects from combining non-invasive brain stimulation with peripheral stimulation techniques. It is likely due to stimulation protocol characteristics, including suboptimal timing of stimulation, overlapping neuroplasticity mechanisms leading to ceiling effects, or insufficient dosing across studies. Other factors that explain this variability could be differences in PwS characteristics (stroke stage, age, or sex) or synergies between stimulation technique mechanisms [48,49]. Optimizing stimulation parameters (specifically, timing, and dosing) in certain stroke populations (acute vs. chronic) could be necessary to unlock their true synergistic benefits in future research.

Regarding the synergistic effects of tDCS and FES, both techniques can enhance corticospinal excitability, with some studies demonstrating increased motor-evoked potentials [50,51,52]. This combined effectiveness might stem from tDCS’s ability to modulate the central nervous system by downregulating activity in the non-affected hemisphere and upregulating the affected hemisphere. This phenomenon follows the interhemispheric inhibition model [53,54] and, coupled with increased activation of the central nervous system areas associated with movement provided by a FES application, could increase the effect of tDCS [55,56]. Additionally, the use of FES and other types of currents can improve motor recruitment and strength in these patients, which could imply peripheral benefits in addition to the central effects of the therapy [57,58,59]. These two forms of stimulation might promote greater neuroplasticity and rehabilitation outcomes through the simultaneous activation of central and peripheral pathways, given that the orthodromic and antidromic signals elicited from tDCS and FES, respectively, could increase the final excitability in **α**-motoneurons [60,61,62].

A standardized montage was observed for tDCS across the included studies, with anodal stimulation consistently applied over the ipsilesional M1 and cathodal placement over the contralesional M1. This interhemispheric setup aligns with the inhibition–excitation model aimed at rebalancing cortical excitability post-stroke [12,13]. Additionally, current densities (ranging from 0.071 [38] to 0.08 mA/cm^2^ [39,40,41,42]), stimulation durations, and session frequencies vary across studies, affecting the degree of cortical modulation. However, significant variability was found in the FES parameters, particularly in pulse width, which ranged from 30 to 500 μs [42]. Other parameters, such as stimulation frequency, intensity, and electrode placement (targeting distal vs. proximal UL muscles), also varied substantially. These inconsistencies could influence the degree of peripheral motor recruitment and, consequently, the overall efficacy of the combined protocol. Future research should aim to standardize or at least systematically explore the effects of various FES parameter configurations, especially pulse width, to better understand their contribution to neuroplasticity and functional recovery in PwS. This review highlights the clinical potential of combining tDCS and FES for UL rehabilitation in PwS. These techniques are safe, easy to implement, and synergistically enhance neuroplasticity, addressing the slow recovery process post-stroke. Their combined effects on cortical excitability and motor recruitment support their integration into clinical practice, with further research needed to refine the protocols and maximize outcomes.

### Limitations

This systematic review has several limitations. The heterogeneity in FES protocols, including variations in intensity, frequency, duration, and targeted muscles, hinders reaching definitive conclusions about optimal combinations with tDCS. The limited number of studies (*n* = 5) and small sample size (*n* = 148) restrict generalizability and reduce statistical power, potentially implementing bias in the findings. Variations in methodological quality, including inconsistencies in randomization, blinding, and control conditions, affect validity despite high inter-rater reliability in bias assessment, which further complicates data synthesis. Additionally, the absence of long-term follow-up limits understanding of sustained effects or potential adverse outcomes. Furthermore, our study does not stratify the sample by the various stages of the pathology, which can have an impact on the effect of the treatment.

Moreover, there are limitations inherent to the review process itself. Although a comprehensive search strategy was employed across multiple databases, there remains the possibility that relevant studies were missed, particularly unpublished or non-English language clinical trials, introducing potential publication and language bias. Additionally, although the bias assessment was performed independently by two reviewers, subjective judgments could still have influenced the results. The limited number of included studies also precluded a meta-analysis, reducing the ability to quantitatively synthesize findings.

Future research requires larger, rigorously designed trials with standardized protocols, consistent metrics, and extended follow-ups to validate the efficacy of tDCS combined with FES for post-stroke rehabilitation. Furthermore, authors should include specific variables to evaluate each ICF domain.

## 5. Conclusions

The combination of tDCS and FES could potentially improve UL motor function in PwS; however, the current evidence is limited and affected by a high risk of bias. There is insufficient evidence to confirm participation-level activity. Therefore, additional high-quality well-designed clinical trials are needed to standardize intervention protocols, optimize stimulation parameters, explore the underlying neurophysiological mechanisms of their synergistic effects, and ensure consistent results across diverse populations. Future research should also prioritize addressing key knowledge gaps, including dose–response relationships and long-term functional outcomes, to better guide clinical implementation.

## Figures and Tables

**Figure 1 neurolint-17-00089-f001:**
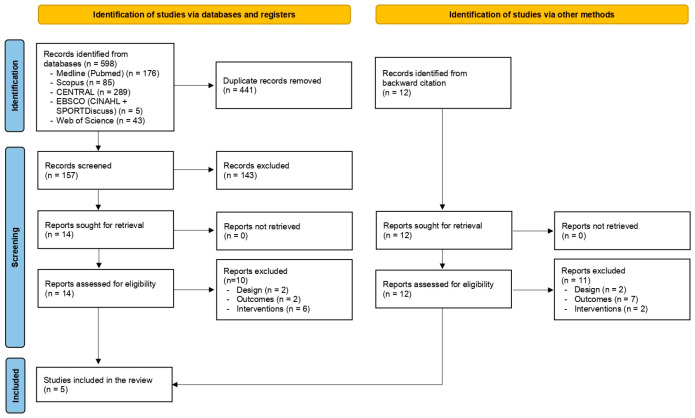
Flowchart of the studies included, following PRISMA guideline recommendations.

**Figure 2 neurolint-17-00089-f002:**
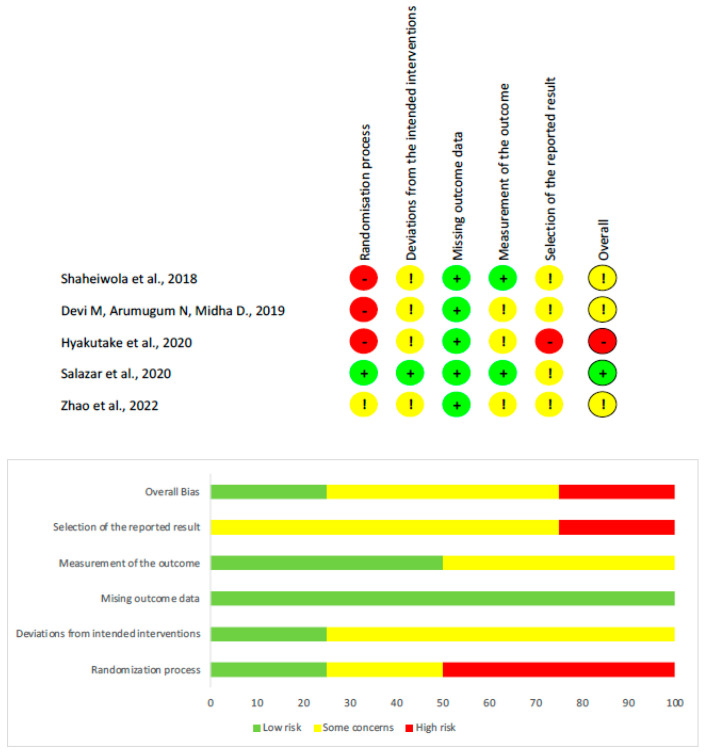
Risk of bias assessment for all studies, categorized by domain [38,39,40,41,42]. Green (+) indicates low risk of bias, yellow (!) indicates some concerns, and red (−) indicates high risk of bias. The lower panel provides a summary representation in percentages for each domain across all studies.

**Table 1 neurolint-17-00089-t001:** Characteristics of the included studies, adding a summary of their results.

Authors, Year	Group, Intervention, Sample (Randomized) [Analyzed]	Sample Size (M/F)	Age (Mean ± SD)	Months/Weeks/Days Since Stroke Onset (Mean ± SD)	Type of Stroke (I/H)	Affected Hemisphere (L/R)	Outcome Measures and Follow-Up	Significant Differences Within and Between Groups
Shaheiwola et al., 2018 [38]	G1: tDCS + FES [15] G2: Sham tDCS + FES [15]	30 (27/3)	50.6 ± 10.14	17 ± 10.26 months	NR	16/14	FMA-UE; WMFT (function and task time); sEMG; amplitude and latency of motor evoked potential. Post.	G1: ↑ FMA-UE, WMFT function, sEMG (anterior deltoid, extensor carpis radialis, and flexor digitorum superficialis). ↓ WFMT task time. G2: ↑ FMA-UE. G1 vs. G2: ↑ FMA-UE, WMFT (function), and general sEMG were favorable to G1.
Devi M, Arumugum N, & Midha D., 2019 [39]	G1: tDCS + FES + Physical therapy [10] G2: tDCS + Physical therapy [10]	20 (NR)	NR	NR	NR	NR	CAHAI, NHPT; FMA-UE; and maximal isometric grip and pinch strength. Post.	G1 and G2: ↑ FMA-UE; CAHAI; hand grip strength; lateral, chuck, and pulp pinch strength. ↓ NHPT. G1 vs. G2: ↑ FMA-UE; CAHAI; Hand grip strength; lateral and chuck pinch; and ↓ NHPT were favorable to G1
Hyakutake et al., 2020 [40]	G1: tDCS + FES + Occupational therapy + BTA [2] G2: tDCS + OT [6]	8 (3/5)	59.5 ± 9.4	54.9 ± 23.2 days	2/6	5/3	FMA-UE; ARAT; MAL. Post-4 months.	G2: ↑ FMA-UE at post, and ↑ MAL (amount of use and quality) at post and 4 months.
Salazar et al., 2020 [41]	G1: tDCS + FES [15] G2: Sham tDCS + FES [15]	30 (20/10)	58 ± 13.43	27.5 ± 12.80 months	25/5	16/14	FMA-UE; Handgrip strength; kinematic analysis of a reaching movement. Post.	G1: ↑ Mean returning and reaching velocity, hand grip strength, FMA-UE. ↓ Movement time, smoothness, trunk forward inclination. G2: ↑ Mean returning velocity, FMA-UE. ↓ Smoothness, trunk forward inclination. G1 vs. G2: ↑ Mean reaching velocity, grip strength, and ↓ movement time were favorable to G1.
Zhao et al., 2022 [42]	G1: tDCS + FES [30] G2: FES [30]	60 (39/21)	61.22 ± 9.44	7.62 ± 2.65 days	NR	26/34	Brunnstrom motor function staging; hemiplegic hand and finger function; and FMA-UE. Post.	G1 and G2: ↑ FMA-UE, functional classification of hemiplegic hand and finger function. ↓ Brunnstrom (hand). G1 vs. G2: ↑ FMA-UE; functional classification of hemiplegic hand and finger function; and ↓ Brunnstrom (hand) were favorable to G1.

Abbreviations: ARAT, Action Research Arm Test; CAHAI, Chedoke Arm and Hand Activity Inventory; BTA, botulinum toxin A; FES, functional electrical stimulation; FMA-UE, Fugl-Meyer Upper Extremity; G, group; MAL, motor activity log; NHPT, Nine-Hole Peg Test; NR, not reported; OT, occupational therapy; sEMG, superficial electromyography; tDCS, transcranial direct current stimulation; WMFT, Wolf Motor Function Test.

**Table 2 neurolint-17-00089-t002:** Transcranial direct current intervention parameters included in each study.

Study	Location of Stimulation	Electrode Size (cm^2^)	Intensity (mA)	Current Density (mA/cm^2^)	Duration of Session (min)
Shaheiwola et al., 2018 [38]	A-ipsilesional M1 C-contralesional M1	25	2	0.08	20
Devi M, Arumugum N, & Midha D., 2019 [39]	NR	NR	1.2	NR	20
Hyakutake et al., 2020 [40]	A-ipsilesional M1 C-contralesional M1	35	2.5	0.071	25
Salazar et al., 2020 [41]	A-ipsilesional M1 C-contralesional M1	25	2	0.08	30
Zhao et al., 2022 [42]	A-ipsilesional M1 C-contralesional M1	NR	2	NR	20

Abbreviations: A, anode; C, cathode; M1, primary motor cortex; NR, not reported.

**Table 3 neurolint-17-00089-t003:** Functional electrical stimulation intervention of each included study.

Study	Frequency	Pulse width	Intensity	Muscles Stimulated	Training with FES
Shaheiwola et al., 2018 [38]	40 Hz	250 µs	The amplitude was selected based on the patient’s needs and was adjusted weekly.	- Anterior deltoid; - Biceps; - Posterior deltoid; - Triceps; - Infraspinatus; - Teres minor; - Teres major; - Extensor carpi radialis longus; - Extensor carpi radialis brevis/ulnaris; - Supinator; - Pronator teres; - Pronator quadratus; - Extensor digitorum; - Abductor pollicis brevis; - Abductor pollicis longus; - Flexor digitorum superficialis; - Flexor digitoris profundus; - Flexor pollicis brevis; - Flexor pollicis longus; - Opponens pollicis.	Moving and placing objects of varying thickness from a jug to the mouth and returning them to their original position. Grasping and inserting small objects using a pincer grip (the last one placed). Performing various movements that combine shoulder rotations, reaching, supination–pronation, and finger extension.
Devi M, Arumugum N, & Midha D., 2019 [39]	20–40 Hz	30–500 μs	Below 100 mA.	Wrist extensors.	Stimulation for 10 min during task-oriented training.
Hyakutake et al., 2020 [40]	35 Hz	50 µs	Contractions without discomfort.	- Wrist extensors; - Finger extensors; - Shoulder flexors.	Activities that include gripping or picking up blocks or pegs, varying in size, using a keyboard, and playing cards.
Salazar et al., 2020 [41]	40 Hz	300 µs	Maximum tolerated.	- Anterior deltoid; - Serratus anterior; - Triceps brachii; - Wrist extensors.	Reaching objects; grasping and releasing.
Zhao et al., 2022 [42]	NR	NR	NR.	Wrist flexors.	Stimulation for 20 min, once per day, 6 days per week, for 2 weeks.

Abbreviations: FES, functional electrical stimulation; NR, not reported.

**Table 4 neurolint-17-00089-t004:** Methodological quality of the selected studies using the PEDro scale.

Study	1	2	3	4	5	6	7	8	9	10	11	Total Score	Quality
Devi M, Arumugum N, & Midha D., 2019 [38]	1	1	0	0	0	0	0	1	1	1	1	5	Fair
Hyakutake et al., 2020 [39]	1	0	0	0	0	0	0	1	1	0	1	3	Poor
Salazar et al., 2020 [40]	1	1	1	1	1	0	1	1	1	1	1	9	Excellent
Shaheiwola et al., 2018 [41]	1	0	0	1	0	0	1	1	1	1	1	6	Good
Zhao et al., 2022 [42]	1	1	0	1	0	0	0	1	1	1	1	6	Good

## Data Availability

Not applicable.

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
