# Peer review of "Combined Transcranial Direct Current Stimulation and Functional Electrical Stimulation for Upper Limbs in Individuals with Stroke: A Systematic Review"

_2035-8377, 2025, doi:10.3390/neurolint17060089_

Round 1
Reviewer 1 Report
Comments and Suggestions for Authors
The manuscript presents a systematic review on the combined use of tDCS and FES for upper limb rehabilitation in stroke patients.
Overall, the topic is clinically relevant, but the review has several methodological and reporting weaknesses which need to be addressed.
To help the authors improve their work, I provide detailed comments and constructive feedback below.
- The abstract is very weak overall. The abstract should mention the search period.
- The abstract results section doesn’t have numerical values and p values.
- The abstract should mention some detailed on patients basic characteristics.
- The conclusion ("moderate evidence") is somewhat vague; it should specify the strength of evidence (e.g., based on GRADE criteria) and highlight the high risk of bias in included studies.
- Keywords are overly broad (e.g., "function," "upper limb"); more specific terms like "motor recovery" or "neurorehabilitation" would improve indexing.
- Introduction is weak too. The rationale for combining tDCS and FES is underdeveloped. While both are discussed separately, the hypothesized synergistic mechanism (e.g., central-peripheral interaction) is not sufficiently explained.
- Add a paragraph explicitly outlining the proposed neurophysiological synergy (e.g., tDCS enhancing cortical plasticity + FES promoting peripheral motor recruitment).
- The gap in literature is mentioned but not critically framed (e.g., no mention of conflicting findings in prior studies).
- Clarify why prior reviews failed to address this combination (e.g., heterogeneity in protocols).
- Inclusion and exclusion criteria should be numbered.
- According to PRISMA, a list of papers which potentially could be included should be mentioned.
- The PICOS framework is used, but the rationale for excluding non-RCTs (e.g., cohort studies) is not justified.
- The PEDro scale is unconventional for systematic reviews (typically used for RCTs); the authors should justify its use over AMSTAR-2 or ROBIS.
- No protocol for handling missing data (e.g., imputation methods) is described.
- Studies in the tables should be arranged chronologically from older to newer or vice versa.
- Table 1 is very confusing and dense. It should be divided to 2 tables.
- The discussion of synergistic effects is speculative (e.g., "likely due to cortical excitability") without direct evidence from included studies.
- The small sample size (n=148) is mentioned, but its impact on statistical power (e.g., risk of Type II error) is not addressed.
- Discussion: Recommendations for tDCS/FES parameters (e.g., electrode placement) are vague.
- The conclusion should be modified as it overstates "promising potential" without acknowledging the high risk of bias in included studies.
- The conclusion doesn’t mention overall approach to prioritize research gaps (e.g., need for dose-response studies).
- The manuscript needs some language and grammar editing.
Author Response
R1:
The abstract is very weak overall. The abstract should mention the search period. The abstract results section doesn’t have numerical values and p values. The abstract should mention some detailed on patients basic characteristics. The conclusion ("moderate evidence") is somewhat vague; it should specify the strength of evidence (e.g., based on GRADE criteria) and highlight the high risk of bias in included studies. Keywords are overly broad (e.g., "function," "upper limb"); more specific terms like "motor recovery" or "neurorehabilitation" would improve indexing.
- Thank you for your valuable comment. We have deeply revised and modified the abstract, including the search period, the main characteristics of the patients. We have also detailed the results obtained in the review highlighting the risk of bias of the selected studies and included more specific terms as keywords.
Introduction is weak too. The rationale for combining tDCS and FES is underdeveloped. While both are discussed separately, the hypothesized synergistic mechanism (e.g., central-peripheral interaction) is not sufficiently explained. Add a paragraph explicitly outlining the proposed neurophysiological synergy (e.g., tDCS enhancing cortical plasticity + FES promoting peripheral motor recruitment).
- Thank you for your comment. We have included a paragraph in introduction section, specifying the posible benefits of combinnig both therapies as a justification for conducting this study
The gap in literature is mentioned but not critically framed (e.g., no mention of conflicting findings in prior studies). Clarify why prior reviews failed to address this combination (e.g., heterogeneity in protocols).
- We addressed these questions as one, and we have changed almost completely the discussion section (paragraph 2 and 3) to fully frame the gap in the literature, including possible causes for difference between NIBS and peripheral stimulation techniques combination.
Inclusion and exclusion criteria should be numbered.
- We have revised the “Selection criteria and data extraction” section to explicitly number both the inclusion and exclusion criteria. This improves clarity and transparency in accordance with the best practices for reporting systematic reviews.
According to PRISMA, a list of papers which potentially could be included should be mentioned.
- We have referenced the studies at the beginning of section 1. Characteristics of the study population and study design.
The PICOS framework is used, but the rationale for excluding non-RCTs (e.g., cohort studies) is not justified.
- We appreciate this comment. Our decision to include only randomized clinical trials (RCTs) and controlled clinical trials with parallel allocation (CCTs) is based on the need to ensure a high level of evidence in a combined intervention - tDCS and FES - which still presents heterogeneous results in the literature. We consider that including observational studies would have introduced greater methodological variability and a substantial risk of bias, thus making it difficult to draw valid conclusions on the effectiveness of the combined treatment. This decision is in line with the methodological recommendations for systematic reviews that seek to establish clinical evidence with the highest degree of internal validity possible. We have clarified this section in the manuscript, explaining in line 144 the following: “The decision to include only randomized controlled trials (RCTs) and controlled clinical trials (CCTs) was based on the need to ensure a high level of internal validity and minimize bias in assessing the combined effects of tDCS and FES”.
The PEDro scale is unconventional for systematic reviews (typically used for RCTs); the authors should justify its use over AMSTAR-2 or ROBIS. (esto se le justifica al revisor, no en el estudio)
- We wish to clarify that the PEDro scale was used exclusively to assess the methodological quality of the randomized clinical trials included in this review, which is consistent with the original purpose of this tool.
However, we recognize that the original phrasing of the manuscript could be misleading. We have therefore corrected line 269 of the manuscript, replacing the ambiguous phrase with a more precise formulation that correctly reflects the purpose and use of the PEDro scale in our study.
No protocol for handling missing data (e.g., imputation methods) is described.
- We appreciate the observation, we agree that a specific statement on this issue was not included. However, we would like to clarify that imputation methods were not applied in the included studies, since the reviewed articles did not report any missing data on the main outcome variables. All qualitative analyses and syntheses were based solely on the data reported by the original studies.
Nevertheless, to enhance the transparency of the manuscript's methodology, we have added an explanatory sentence to the 'Risk of bias assessment' section to this effect:
'No imputation methods were required, as none of the included studies reported missing data for the primary outcome measures.'
Studies in the tables should be arranged chronologically from older to newer or vice versa.
Thank you for your suggestion regarding the order of the studies included in the tables.
We have reorganized the corresponding tables to present the studies in ascending chronological order (from the oldest to the most recent), as recommended. However, in the case of Table 4 (PEDro methodology) and the risk of bias assessment figure (RoB2), we have opted to maintain an alphabetical order by first author, with the aim of facilitating the cross-referencing of studies in other sections of the manuscript.
Table 1 is very confusing and dense. It should be divided to 2 tables.
After carefully evaluating your suggestion, we have opted to keep the information in a single table, since we consider that dividing it into two could make it difficult to trace the data between the characteristics of the studies and the parameters used in the interventions. Likewise, such a division would unnecessarily increase the number of tables in the manuscript without substantially improving the comprehensibility of the content.
Nevertheless, we have revised the table design to optimize its clarity and readability, ensuring a cleaner and more accessible presentation of the information.
We thank you again for your suggestion, which has allowed us to reflect on the most appropriate presentation of the data.
The discussion of synergistic effects is speculative (e.g., "likely due to cortical excitability") without direct evidence from included studies.
- We have changed and added more information to clarify that point in the fourth paragraph of the discussion and referencing a meta-analysis in the first paragraph “Meta-Analysis of Functional Electrical Stimulation Combined with Occupational Therapy on Post-Stroke Limb Functional Recovery and Quality of Life” Mei Sunb Che Jiang Jianbo Zhangc Kuihong; Cheng, Wei; Zheng Xiwu; Zhang Xiaon; Wuc Zhuang; Chenc Gaoquan Luo; Gang Zhao. We hope this reference and new discussion section could clarify the question of the reviewers.
The small sample size (n=148) is mentioned, but its impact on statistical power (e.g., risk of Type II error) is not addressed.
Since our study is a systematic review without quantitative synthesis (i.e., no meta-analysis), we did not perform formal statistical power calculations. However, we acknowledge that the small number of included studies and total participants (n=148) limits the strength of the evidence and may increase the risk of Type II error — that is, the inability to detect true effects due to insufficient sample size. We have reflected this limitation in the discussion, noting that the small evidence base reduces the certainty of our conclusions and highlights the need for further research.
Discussion: Recommendations for tDCS/FES parameters (e.g., electrode placement) are vague.
We have clarified this aspect by adding a new paragraph in the Discussion section. In it, we specify that all included studies employed a consistent tDCS montage, with the anode placed over the ipsilesional M1 and the cathode over the contralesional M1, in accordance with the interhemispheric inhibition model. Regarding FES, we now highlight the substantial variability in stimulation parameters across studies, particularly pulse width, as well as electrode placement targeting different upper limb muscles. We also comment on how this variability may affect outcomes and advocate for future research to explore and optimize these parameters in a systematic manner to improve clinical applicability.
The conclusion should be modified as it overstates "promising potential" without acknowledging the high risk of bias in included studies. The conclusion doesn’t mention overall approach to prioritize research gaps (e.g., need for dose-response studies).
We agree that the original phrasing may have overstated the strength of the evidence given the high risk of bias in the included studies. We have revised the conclusion to better reflect these limitations and now emphasize the need for rigorous future research. We also incorporated your suggestion to highlight research priorities, such as dose-response studies and long-term outcomes.
The manuscript needs some language and grammar editing.
Thank you for your suggestion, we sent the manuscript to a professional english edition service, to improve grammar and language used.

Reviewer 2 Report
Comments and Suggestions for Authors
It is a really important and well written study. It followed most of PRISMA checklist points. Below some minor suggestion that can clarify come point.
Please double check abbreviations of upper limb (UL), it uses must be consistent throughout all manuscript.
Abstract
In methods, add more information about inclusion criteria as year of study publication and if participants with acute and/or chronic stroke were considered.
In methods, cite studies Qualitative Analysis conducted.
IN results add information about studies quality
Methods
Does time since stroke was considered in inclusion criteria? This point can directly influence studies results. If this point was not considered, please comment in discussion section.
Please highlight which terms used for the search strategy were MESH terms
Does automation tools used in the selection process?
Discussion
Add reasons that may justify differences between the results reported here and cited previous study’s results.
Add reasons that can justify differences between results of selected studies.
In limitations, discuss any limitations of the review processes used. Was cited just selected studies’ limitations.
Author Response
R2: Comments and Suggestions for Authors
Please double check abbreviations of upper limb (UL), it uses must be consistent throughout all manuscript.
- We have revised the UL acronym for Upper Limb throughout the manuscript and optimized its use, to preserve the comprehensibility of the manuscript.
Abstract
In methods, add more information about inclusion criteria as year of study publication and if participants with acute and/or chronic stroke were considered. In methods, cite studies Qualitative Analysis conducted.
- We mentioned the Qualitative Analysis conducted in the abstract, which was completely rewritten.
Besides, we have rewritten the “Population” section in methods: “The participants included in the reviewed studies were PwS, aged over 18 years, and presenting UL deficits, regardless of the time since stroke onset (acute, subacute, or chronic).”
Regarding qualitative analysis, we included several references that used the 5 grades system to evaluate the qualitative analysis.
In results add information about studies quality
- We have rewritten the section “3.4 Methodological quality and risk of bias results” in a clearer way to report a more precise information concerning studies quality.
Methods
Does time since stroke was considered in inclusion criteria? This point can directly influence studies results. If this point was not considered, please comment in discussion section.
- Thank you for your valuable comment, we are completely agree with this issue, so we have included it in “limitations” section
Please highlight which terms used for the search strategy were MESH terms
- We have clarified it in the search strategy section: “The Mesh terms used for the search strategy were: “transcranial direct current stimulation”, “electrical stimulation therapy”, “paresis”, “stroke”, and “upper extremity”.”
Does automation tools used in the selection process?
- We added it at the end of selection criteria and data extraction.
Discussion
Add reasons that may justify differences between the results reported here and cited previous study’s results.
- We have changed almost completely the discussion section (paragraph 2, 3 and 4), including possible causes for difference between our results and other studies that combined NIBS and peripheral stimulation techniques.
Add reasons that can justify differences between results of selected studies.
- We added a new paragraph in discussion (middle and end part of paragraph 2). We hope your question is answered with our explanation.
In limitations, discuss any limitations of the review processes used. Was cited just selected studies’ limitations.
- We have now added a paragraph addressing potential limitations of the systematic review methodology, including risks of publication and language bias, limitations related to data synthesis, and possible subjectivity in risk of bias assessment.

Round 2
Reviewer 1 Report
Comments and Suggestions for Authors
The authors have addressed my comments well.